# Injury Occurrence and Return to Dance in Professional Ballet: Prospective Analysis of Specific Correlates

**DOI:** 10.3390/ijerph16050765

**Published:** 2019-03-03

**Authors:** Bozidar Novosel, Damir Sekulic, Mia Peric, Miran Kondric, Petra Zaletel

**Affiliations:** 1Department of Orthopedics and Traumatology, General Hospital Varazdin, Varazdin 42000, Croatia; bozidar.novosel@gmail.com; 2Faculty of Medicine, University of Mostar, Mostar 88000, Bosnia and Herzegovina; 3Faculty of Kinesiology, University of Split, Split 21000, Croatia; mia.peric@kifst.hr; 4Faculty of Sport, University of Ljubljana, Ljubljana 1000, Slovenia; miran.kondric@fsp.uni-lj.si (M.K.); petra.zaletel@fsp.uni-lj.si (P.Z.)

**Keywords:** dance, prevalence of injury, time-off, predictors, AUDIT

## Abstract

Professional ballet is a highly challenging art, but studies have rarely examined factors associated with injury status in ballet professionals. This study aimed to prospectively examine gender-specific correlates of injury occurrence and time-off from injury in professional ballet dancers over a one-year period. The participants were 99 professional ballet dancers (41 males and 58 females). Variables included: (i) predictors: sociodemographic data (age, educational status), ballet-related factors (i.e., experience in ballet, ballet status), cigarette smoking, alcohol drinking, and consumption of illicit drugs; and (ii) outcomes: injury occurrence and time-off from injury. Participants were questioned on predictors at the beginning of the season, while data on outcomes were collected continuously once per month over the study period. Dancers reported total of 196 injuries (1.9 injuries (95% CI: 1.6–2.3) per dancer in average), corresponding to 1.4 injuries per 1000 dance-hours (95% CI: 1.1–1.7). In females, cigarette smoking was a predictor of injury occurrence in females (OR: 4.33, 95% CI: 1.05–17.85). Alcohol drinking was a risk factor for absence from dance in females (OR: 1.29, 95% CI: 1.01–4.21) and males (OR: 1.21, 95% CI: 1.05–3.41). Less experienced dancers were more absent from dance as a result of injury than their more experienced peers (Mann-Whitney Z: 2.02, *p* < 0.04). Ballet dancers and their managers should be aware of the findings of this study to make informed decisions on their behavior (dancers) or to initiate specific programs aimed at the prevention of substance use and misuse in this profession (managers).

## 1. Introduction

Professional ballet dance is a highly demanding performance art [1,2,3]. Given the high physiological and psychological stress, injuries are common in ballet. Traumatic injuries (e.g., injuries that occur as a result of acute stress) are less common and are mostly associated with loss of balance during the practice or performance (e.g., ankle strain, hamstring strain, patella dislocation). More common injuries occur as a result of systematic overload (i.e., overuse injuries), and involve injuries of the lower extremities and lower back [4,5,6]. Mechanisms of injury are well known [4,7]. In brief, professional ballet dancers train up to 40 h per week in addition to performances, and the high number of training hours and repetition results in gradual wear and tear that progressively worsens over time [8]. In other words, ballet dancers exploit their locomotor system, which leads to breaking the limits of the adaptive mechanisms and results in motor system dysfunctions and injury.

Previous studies have reported the injury occurrence and some factors associated with injury occurrence in ballet, but only few examined professional dancers [9,10]. In a study of young dancers (10 to 18 years of age) involved in UK Centres for Advanced Training in dance (CATs; including ballet dancers), the authors identified intensity of training (such as number of hours of training per week) as an important risk factor for injury [10]. Australian authors prospectively examined injuries in pre-professional ballet dancers (15–19 years of age) and showed an injury rate of 1.38/1000 h of dance [9]. Further, the ankle was the most commonly injured location, while knee injuries were associated with the greatest time loss per injury. In a study of professional ballet dancers, authors assessed the incidence and severity of injuries over a 1-year period [6]. In addition to injury (approximately 4.1/1000 h), the authors reported severity of injuries (i.e., absence from activity as a result of injury), with a mean severity of 3 days for females and 9 days for males [6]. Indeed, professional ballet dancers directly depend on their ability to perform. Therefore, severity of injury is an important factor of their overall success and achievement. 

With the growing interest of injury in ballet, there is an evident increase in studies that examined factors associated with injury prevalence in this profession [11,12]. Studies performed on ballet students investigated body composition as a predictor of time-off from injury and reported body fat as significantly correlated with the time-off from acute injury (*r* = −0.32, *p* = 0.04) and the total time off from any injury (*r* = −0.31, *p* = 0.048) [13]. More recent investigations identified factors of body build (e.g., somatotype components) related to specifically located injuries in pre-professional female ballet dancers [5]. Briefly, endomorphy was related to ankle-injury, ectomorphy to foot injury, and body-mass to injury to the toes [5]. With regard to functional variables, studies reported poor lumbopelvic movement control, inappropriate transversus abdominis contraction, decreased lower-extremity strength, and poor aerobic endurance as important correlates of injury occurrence [12,14,15]. Meanwhile, joint range of motion was not a significant predictor of injury in pre-professional ballet dancers [16]. 

In their culture of excellence, ballet dancers are committed to achieve high performing standards but regularly must reach societal standards of aesthetics (i.e., lean and elegant body figure), which results in high physical and emotional stress [17]. Not surprisingly, previous studies reported disturbing figures of substance use and misuse (SUM) in this profession [1,2,18]. A study performed on professional ballet dancers reported a high prevalence of binge alcohol drinking in male dancers (greater than two-thirds of males reported binging at least once per week) and smoking in females (>30% reported daily smoking, while 20% smoked more than a pack per day) [18]. Comparative analysis of different performing arts (sport dancers, ballet and synchronized swimmers) revealed a high positive tendency toward performance enhancers (i.e., doping behavior) in ballet dancers [2], while consumption of the appetite suppressants and pain killers is also known as an important problem in this profession [1]. Interestingly, there is an evident lack of information about possible association between SUM and injury status in ballet, although studies in other high physically and emotionally demanding activities and professions (i.e., competitive sports, military professionals) frequently reported SUM as a factor that negatively influences both injury occurrence and recovery [19,20,21,22].

Despite the significant injury problem in ballet, there is a lack of information on factors that may be specifically associated with injury occurrence and injury severity in professional ballet dancers. Therefore, the primary aim of this prospective study was to examine some specific correlates of injury occurrence and factors associated with time loss as a consequence of injury in male and female professional ballet dancers. Initially, we hypothesized that the studied sociodemographic, ballet-specific and SUM factors will be specifically correlated to injury occurrence and time-loss (i.e., severity of injury) in studied ballet dancers. Identification of risk factors will allow for developing the accurate intervention strategies to mitigate the injury problem in this profession.

## 2. Materials and Methods 

### 2.1. Participants

The participants in this one-year prospective study were 99 professional ballet dancers (41 males and 58 females). All participants were members of National Ballet Theaters from Croatia (*n* = 67; 39 females) and Slovenia (*n* = 32; 19 females) and were observed during the one-year period (starting from early September). The invitation for a participation was individually sent to all professionals of four ballet theatres (two of two, and two of four ballet theatres in Slovenia and Croatia, respectively). The target population was 175 dancers (70 and 85 in Slovenia and Croatia, respectively). Therefore, the response rate was 61% (43% and 79% for Slovenia and Croatia, respectively). Different ballet groups were observed during different years but all were tested between 2016 and 2018. During the course of the study, most of them participated in 25–30 dance hours per week in addition to performance. The Institutional Ethical Board of the corresponding author reviewed and approved the investigation (No. 2181-205-02-05-14-004). The participants were informed that they could refuse participation and withdraw from the study at any time for any reason, and their informed written consent was obtained. Participation was anonymous, and no personal data directly connected to an individual were included in the study. For the purpose of following responses over repeated measures, dancers used anonymous identification code (i.e., they were instructed to use the last three digits of their e-mail password to remember it easily). 

### 2.2. Variables and Measurement

Variables in this study were collected by previously used and validated questionnaires [1,2,5] and included sociodemographic data, ballet-related factors, SUM data (predictors), and data about injuries (outcomes). Participants were questioned on predictors at the beginning of the season. Data about outcomes were collected continuously over the next season (year) once per month. Given that we observed variables of SUM, we deemed anonymity of the participants as a particularly important issue, and all questions that could be possibly connected to single person were categorized (i.e., age was not asked in real number of years but in age-categories; see later for details). 

Sociodemographic factors observed were gender (male – female), age (obtained on a scale consisting of seven possible answers/categories (“<19 years”, “19–22”; “23–26; “27–30”, “31–34”, “35–38”, “+38 years”), and highest educational level achieved (including “elementary school”, “high-school”, “college/university student”, and “college/university level”). Ballet-related factors were observed to identify participants’ status and level achieved in ballet and included questions about experience in ballet (“10–15 years”, “16–20 years”, “21–25 years” and “>25 years”), the current ballet level/performance (“Principal”-Soloist”-First artist”-Corps de ballet”), and number of training (dance) hours per week in addition to performance (“<20 h”, “21–25 h”, “26–30 h”, “>30 h per week”). The SUM was evidenced by questions on consumption of cigarettes, alcohol and illicit drugs. Cigarette smoking was assessed on a scale with the following responses: (“Never smoked”, “Quit”, “From time to time, but not daily”, “Less than 10 cigarettes daily”, “10–20 cigarettes daily”, “From one to two packs daily”, “More than 2 packs daily”). Participants were later classified as nonsmokers (those who responded “No” and “Quit”) or smokers (remaining answers). The variables regarding alcohol consumption included two questions. The first one evidenced binge drinking practice on a scale consisting of seven possible answers (“I don’t drink alcohol”, “I drink alcohol but never binge”, “Rarely”, “Binge drinking couple of times per year”, “Binge drinking once a month or so” , “Binging once a week”, “Binging couple of times per week”), while the second was the standardized Alcohol Use Disorders Identification Test (AUDIT) proposed by World Health Organization consisting of 10 items with scores ranging from 0 to 4, resulting in a scale of 0 to 40 [23,24]. The scale for drug consumption included questions about the consumption of marijuana, hashish, heroin, cocaine, sedatives, and most party drugs (e.g., ecstasy, amphetamines). Participants were later categorized as “drug users” and “nonusers” [25].

Injury status was observed by questions on injury occurrence, location of injury, and severity of injury. Following the suggestions from previous studies, an injury was defined as “any physical complaint sustained as a result of performance or training, irrespective of the need for medical attention or time lost from activity” [5,26]. Dancers were asked to report the injury occurrence on 15 body locations (head, neck, shoulder, chest, elbow, wrist, upper back, lower back, hips, gluteus, knee, calf, Achilles’ tendon, ankle and foot), and the absence from ballet practice as a result of injury (responded on a six point scale, including “No absence at all”, “Less than 3 days”, “4–7 days”, “>7 days”). For the purpose of statistical analyses (see later for details), the participants were later grouped according to injury status (injured vs. non-injured) and injury severity (absent vs. non-absent from dance/performance). 

### 2.3. Statistics

The statistics were reported as frequencies (number of observations) and percentage (Appendix A). Injury rates were calculated as the ratio of injuries per 1000 h of dance (HD) with corresponding 95% confidence intervals (95% CI). The differences between genders were evidenced by Mann-Whitney test (for number of injuries), and Chi square test (injury occurrence). Additionally, to establish the difference in injury rates between genders the Odds Ratio (OR) with corresponding 95% CI were calculated. 

To establish the associations between AUDIT, cigarette smoking (smokers vs. non-smokers) and consumption of illicit drugs (users vs. non-users), and criteria, logistic regression analyses were performed for binomial outcomes (injury occurrence: injured vs. non-injured; time-off from injury: absent vs. non-absent). To establish the association between remaining predictors (age, educational level, ballet related factors (experience, performance level, number of training hours), cigarette smoking (observed at ordinal scale), and binge drinking), and two outcomes (injury occurrence, time-off from injury) we calculated Mann-Whitney test. A value of *p* < 0.05 was considered statistically significant. Statistical analyses were performed using Statistica ver 13.3 (Tibco Software Inc., Palo Alto, CA, USA).

## 3. Results

During the one-year period, a total of 196 injuries occurred, equating to an injury rate of 1.9 injuries per dancer per year (95% CI: 1.6–2.3). The higher frequency is founded in males than in females (2.4 (95% CI: 1.7–3.1) and 1.59 (95% CI: 1.1–2.1) injuries per dancer per year in males and females, respectively). It altogether corresponds with 1.4 (95% CI: 1.1–1.7) injuries per 1000 dance hours (1.7 (95% CI: 1.2–2.2) and 1.1 (95% CI: 0.7–1.5) injuries per 1000 HD in males and females, respectively).

The percentage of dancers who reported no injury was similar across genders (23.7% and 25.42% for males and females, respectively), with no significant difference between genders (Chi square: 0.20, *p* = 0.66; OR: 1.55 (95% CI: 0.81–2.94)). Also, there was no significant difference between genders when number of injuries was observed on an ordinal scale (Mann-Whitney Z value: 1.11, *p* = 0.26). 

The ankle was the most common injured location in females, accounting for 36.5% of all injuries, followed by calf (14.6% of all injuries), foot and knee (both accounting for 12.5% of all injuries in females). Male dancers most commonly injured lower back (20% of all reported injuries) followed by ankle (14%), knee (11%) and foot (10% of all reported injuries in males) (Table 1). 

In addition, 38% of all injuries required no single day of rest (absence from dance practice), with some difference between genders. Specifically, 32% of injuries that occurred in males and 42% of injuries in females did not result in absence from training/performance. However, when genders were compared for severity of injuries, no significant difference was noted between male and female dancers (Mann-Whitney Z value: 0.44, *p* = 0.65). 

In females, cigarette smoking was associated with injury occurrence with a higher likelihood of injury in those dancers who smoke (OR: 4.33, 95% CI: 1.05–17.85). Alcohol drinking observed by the AUDIT scale was a significant predictor of absence from dance in females (OR: 1.29, 95% CI: 1.01–4.21) and males (OR: 1.21, 95% CI: 1.05–3.41) (Table 2).

Mann-Whitney test revealed significant differences between injured and non-injured female dancers for cigarette smoking status observed at ordinal scale (Z: 2.05, *p* = 0.04), with higher rates of smoking in injured dancers (55% and 41% of non-smokers among non-injured and injured dancers, respectively). Finally, less experienced dancers were more likely to be absent from dance as a result of injury than their more experienced peers (Z: 2.02, *p* < 0.04; with 39% and 30% of dancers with >25 years of experience in ballet among non-absent and absent, respectively) (Table 3). 

## 4. Discussion

The results of the study revealed several important findings. First, higher risk of injury was evidenced in female dancers who smoke cigarettes. Second, alcohol drinking measured by the AUDIT scale was related to absence from dance as a result of injury both in males and females, while experience in ballet was negatively related to absence from dance in females. Therefore, our initial study hypothesis was confirmed. Before discussing those results directly related to our study aims, we will overview findings on injury prevalence in studied dancers. 

### 4.1. Injury Prevalence in Ballet Dancers

According to previous reports, injury-rates in ballet vary [6,9,16]. In our study, the total number of 193 injuries that occurred over a one-year period of observation corresponds with 1.4 injuries per 1000 h of dance (HD). This figure is somewhat higher than the injury rate reported for young ballet dancers (0.77 injuries/1000 HD) [16]. Meanwhile, our results corresponds with a report on Australian pre-professionals (15–19 years of age) where authors reported an injury rate of 1.38/100 HD [9] but are much lower than the injury rate reported in a study that investigated a UK professional ballet company (4.4 injuries/1000 HD) [6]. 

Specifically, in the study where authors reported three-fold increased rates of injury than we found herein, dancers were members of a professional company which performed “on tour” [6]. Half of the company’s performances occurred in its home theater and half on tour; therefore, the company did not have a consistent touring surface. Additionally, company dancers averaged more than 35 h per week [6]. Meanwhile, our dancers were located at home theaters, performed more than 90% of dance hours on consistent and familiar dancing surfaces, and averaged 25–30 h per week. These factors may collectively represent possible causes for difference in injury rates.

We noted evident gender-differences in the injured body-region. Briefly, females mostly injured ankles (36%), while lower back was the most common injured region in males, accounting for 20% of all injuries. Similarly, in the study on Australian pre-professionals, authors reported ankle as the most commonly injured location (25% of all injuries) [9], and this finding is in accordance with reports from UK Centers for Advanced Training [10]. Interestingly, in the previously cited study on the ballet professional company, authors reported highest prevalence of lower leg injuries (i.e., calf injuries) both in males and females [6], which indirectly confirms our previous discussion on inconsistency in dancing surfaces as a risk factor for injury occurrence [27].

### 4.2. Cigarette Smoking and Injury Occurrence in Female Ballet Dancers

Studies repeatedly reported positive correlations between cigarette smoking and injury occurrence in investigations involving military personal and athletes [19,20,21]. In the early works on a problem, authors offered specific sociopsychological explanations of such relationships (i.e., smokers were more likely to be “risk takers” and therefore more involved in “accidents” which consequently led to injury) [28]. However, later prospective and longitudinal studies evidenced clear physiological background of the negative influence of smoking on musculoskeletal health, which probably explains our findings on higher injury rates in female ballet dancers as well. 

First, cigarette smoking has anti-estrogenic effects (i.e., reduces estrogen level) by different mechanisms, such as inhibiting the enzyme aromatase, increasing 2-α-hydroxylation with irreversible conversion of estrone to the inactive metabolite 2-methoxyestrone, and increasing sex-hormone binding globulin levels with subsequent reductions in active free estradiol [29]. Additionally, some systematic effects of cigarette smoking, such as chronic inflammation, vascular damage, and increased oxidative stress, may negatively affect bone mineral content [29,30]. It altogether alters bone health and results in a higher injury risk, especially in those professions and activities where high intensity (load) is placed on bones. 

With regard to ballet dancers, the association between cigarette smoking and health of connective tissues, specifically tendons, is interesting. Experimental studies confirmed correlation between smoking and shoulder pain and tendinopathy (the greater the smoking the more pain and tendinopathy) as well as increased risk of biceps tendon rupture [31,32,33,34]. Therefore, it seems reasonable to expect similar negative influence of smoking on other tendon structures (i.e., Achilles’ tendon in ballet dancers). 

### 4.3. Experience in Ballet and Injury Status

Our results showed that more experienced but not necessarily older (i.e., there was no association between age and injury status) female ballet dancers are less likely to be out of dance after they suffered injury. Several explanations are plausible to explain these findings. First, it is possible that more experienced dancers are able to tolerate pain and discomfort more effectively than their less experienced colleagues [35]. Second, it might be supposed that more experienced dancers are stressed to maintain their status in ballet and therefore avoid longer recovery (i.e., they are reluctant to report injuries for fear of being recognized as “unreliable” or losing their roles) [35].

Indeed, the professional career of ballet dancers is short, but reports show that dancers do not have a realistic idea of how long their career would last [36]. The “aDvANCE” project conducted in the early 2000s identified health-problems as the leading reason of career end in dancers, but it is interesting that approximately 50% of dancers (current and former) self-perceived “sense of emptiness” as the most important challenge in their post-performance career transition [37]. 

When the challenges that lie ahead of dancers at the end of their professional career were investigated, previous studies underscored the notion that the end of a career in dance is recognized as certain “life’s little death”, with dancers having to mourn the loss before embarking on a new career [36]. Bearing that in mind, it is understandable that dancers will try to prolong the end of their dancing career as much as possible, and such intentions are logically more prevalent in more experienced dancers. Pushing the body over the physiological limits is a logical consequence of such attitude.

### 4.4. Alcohol Consumption and Time-off as a Result of Injury

Alcohol consumption measured by the AUDIT scale was negatively related to time-off from injury both in males and females. To the best of our knowledge, no study to date reported associations between alcohol consumption and injury, which is surprising mostly due to two reasons. First, previous reports evidenced disturbing prevalence of alcohol consumption in ballet; and second, drinking is related to injury status in physically demanding professions, such as sports and the army [20,38]. Therefore, our results of the negative influence of alcohol consumption on recovery from injury in ballet are pioneering to some extent, and a brief overview of the physiological mechanisms that generate such associations is provided.

In ballet, as in any other physically demanding activities optimal recovery from damage to skeletal muscle is imperative. It is well established that the likelihood of further injury increases if full recovery is not achieved [19]. In normal circumstances, the immune system responds to trauma by initiating specific inflammatory responses. Many of these responses are altered by alcohol molecules [39,40].

Next, one of the key factors of recovery from injury is status of anabolic hormones, while alcohol detrimentally impacts normal hormonal balance, including the balance of anabolic hormones [41]. Additionally, it must not be ignored that consumption of alcohol may be associated with decreased duration and quality of sleep [42], which is another important factor in recovery. Further, alcohol is a diuretic and increases dehydration, which is highly important factor in overall metabolism and consequent recovery from injury [43,44,45]. Therefore, the fact that alcohol drinking is related to time-off in dancers who consume alcohol to a greater extent deserves particular attention. 

### 4.5. Limitations and Strengths

The first limitation of the study is related to relative non-accuracy of data gathering, since some variables were not collected on most adequate scales (i.e., age was not reported in exact numbers but in 3–4 year spans). However, this was necessary to assure anonymity of the participants, which is absolutely crucial since the study included data on SUM. Further, dancers may lean toward socially acceptable answers (e.g., report lower SUM). Meanwhile, we believe that the tendency toward social desirability was low because all dancers were adults, and we tried to preserve their anonymity throughout specific study design (i.e., type of the questionnaire, usage of the identification codes). Additionally, the study included professionals from several ballet troops; therefore, they varied in overall training and performance (i.e., different numbers and levels of performances), while different troops were observed throughout different years. This limitation almost certainly limits the generalizability of the results to some extent. Finally, some of the observed variables were dichotomized, which certainly reduced the possibility to observe different gradients of variables and its influence on criterion. 

This is one of the first studies that observed SUM variables as possible predictors of injury status in professional ballet. Further, an important strength of the investigation is the prospective study design, which allowed the identification of the cause-effect relationship between observed predictors and outcomes. Next, gender-specific analyses allowed precise identification of the studied problem. Finally, the study included ballet professionals with significant experience in ballet and was one of the first that explored factors related to their return to work (dance) after an injury. Knowing the high demands and short-lived career of ballet dancers, we believe that our study, although not the final word on a problem, will increase knowledge in the field. 

## 5. Conclusions

Our results confirm the necessity of further evaluation of the associations between SUM and overall health status, including injury status in ballet dance. Namely, herein we evidenced highly specific associations between cigarette smoking and alcohol drinking with injury occurrence and time-off from injury in ballet dancers, which should be investigated in more detail. For such a purpose, it would be particularly important to study eventual covariates of SUM itself (i.e., appetite-suppressing effects of smoking and psychoactive relaxing effects of alcohol drinking). The identification of such associations will probably allow us to identify the true background of both SUM behavior and the influence of SUM on injury status in this high demanding profession.

Dancers and their managers should be informed of evidenced negative associations between SUM and injury status. Since dancers directly depend on the quality of the overall health status, it is reasonable to expect that at least some of them will objectively evaluate the here evidenced influence of SUM on their body and will be able to draw informed decisions on their future behavior. On the other hand, based on such information, responsible ballet managers will be able to make certain efforts toward the prevention of SUM in their troops. Neither dancers nor managers may consider alcohol and cigarettes to be harmful in the same way as they consider illicit drugs to be. Therefore, to provide a full understanding of the implications alcohol and cigarettes may have on dancing performance, recovery and, perhaps most importantly, general health is required. 

This study confirmed the association between specific ballet factors (e.g., experience in ballet) and time-off from injury in female dancers. Although we were not able to discuss this relationship in more detail, there is a certain possibility that the associations are generated by solicitude for their future career in more experienced dancers. Therefore, future qualitative studies on the problem are warranted. 

The variables included in such qualitative investigations should not include only those factors observed herein. In brief, according to our results it is clear that due to need for constant achievement and excellence in this profession, different socio-environmental stressors must be observed. Therefore, factors such as level of outside support, level of competition among dancers, etc. must be observed as well. Authors of the study believe that conducting studies which will allow identification of those indices (by face to face interviews, focus groups) will allow identification of different factors related not only to injury-status, but also, to overall health-status and well-being of ballet dancers.

## Figures and Tables

**Table 1 ijerph-16-00765-t001:** Injury locations in ballet dancers (*n*—number of observations, %—percentage).

Locations	Total		Males		Females	
	*n*	%	*n*	%	*n*	%
Head	0	0.0%	0	0.0%	0	0.0%
Neck	8	4.1%	5	5.0%	3	3.1%
Shoulder	8	4.1%	7	7.0%	1	1.0%
Chest	2	1.0%	2	2.0%	0	0.0%
Elbow	2	1.0%	2	2.0%	0	0.0%
Wrist	1	0.5%	1	1.0%	0	0.0%
Upper back	9	4.6%	5	5.0%	4	4.2%
Lower back	31	15.8%	20	20.0%	11	11.5%
Hips	7	3.6%	7	7.0%	0	0.0%
Gluteus	5	2.6%	5	5.0%	0	0.0%
Knee	23	11.7%	11	11.0%	12	12.5%
Calf	21	10.7%	7	7.0%	14	14.6%
Achilles’ Tendon	8	4.1%	4	4.0%	4	4.2%
Ankle	49	25.0%	14	14.0%	35	36.5%
Foot	22	11.2%	10	10.0%	12	12.5%
Total	196		100		96	

**Table 2 ijerph-16-00765-t002:** Results of the logistic regression for the injury occurrence and time of from injury.

Factors	Injury Occurrence				Time off from Injury			
	MALES (*n* = 41)		FEMALES (*n* = 58)		MALES (*n* = 41)		FEMALES (*n* = 58)	
	OR	95% CI	OR	95% CI	OR	95% CI	OR	95% CI
Cigarette smoking								
Yes	1.5	0.40–5.65	4.33	1.05–17.85	0.97	0.71–1.33	1.9	0.66–5.42
No	REF				REF		REF	
AUDIT ^CONT^	1.15	0.91–1.48	0.88	0.66–1.17	1.21	1.05–3.41	1.29	1.01–4.21
Consumption of illicit drugs								
Yes	0.92	0.25–3.39	0.99	0.66–1.57	0.75	0.20–2.85	1.07	0.34–3.36
No	REF		REF		REF		REF	

Legend: ^CONT^ denotes continuous variable, REF—denotes referent value, AUDIT—Alcohol Use Disorder Identification Test.

**Table 3 ijerph-16-00765-t003:** Results of the Mann-Whitney test (Z—Z value; *p*—level of significance) for injury occurrence and time of from injury, in male and female ballet dancers.

Factors	Injury Occurrence (Yes-No)		Time-Off from Injury (Yes-No)	
	Males	Females	Males	Females
	Z (*p*)	Z (*p*)	Z (*p*)	Z (*p*)
Age	0.60 (0.54)	0.58 (0.56)	0.69 (0.48)	0.30 (0.75)
Educational level	0.27 (0.78)	0.78 (0.42)	0.56 (0.56)	1.30 (0.19)
Experience in ballet	1.16 (0.26)	1.27 (0.20)	0.55 (0.57)	2.02 (0.04)
Ballet performance level	0.99 (0.31)	0.46 (0.63)	0.26 (0.79)	0.80 (0.42)
Number of training hours	1.18 (0.23)	1.24 (0.21)	1.90 (0.06)	0.39 (0.69)
Binge drinking	0.45 (0.65)	0.67 (0.49)	1.69 (0.09)	1.86 (0.06)
Cigarette smoking ^ordinal^	0.87 (0.38)	2.05 (0.04)	0.60 (0.54)	0.73 (0.46)

Legend: ^ordinal^ denotes “cigarette smoking” observed as ordinal variable with seven possible responses.

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
