# Peer review of "Injury Occurrence and Return to Dance in Professional Ballet: Prospective Analysis of Specific Correlates"

_ijerph, 2019, doi:10.3390/ijerph16050765_

Round 1

Reviewer 1 Report

Review of “Injury Occurrence and return to dance in professional ballet:  gender specific prospective analysis of specific correlates.”

The manuscript, “Injury Occurrence and return to dance in professional ballet:  gender specific prospective analysis of specific correlates” adds to the injury prevention literature in the area of dance and especially ballet.  My major and minor comments are below.

My first comment has to do with the title of the manuscript.  While comparisons were made between males and females throughout the manuscript, I do not think it needs to be in the title as through most of the manuscript dancers as a whole were considered.

The Introduction provided an interesting overview of dance injuries and the lack of information available- especially in ballet.   While it is understandable why the variables such as substance use and smoking were reviewed, there was no information provided about what other variables might be considered in this research-including environmental variables that may play a role.  These variables might include socio-environmental stressors (need for constant achievement and excellence), lack of outside support, competition among dancers, etc.  These types of variables might best be analyzed through more qualitative techniques to help better explain why ballet dancers feel the need to use substances and smoke and perform other health-comprising behaviors.  While it is realized that not all variables could possibly be assessed, statements regarding other factors that may play a role should be provided.

For the Materials and Methods section, it was not clear when the year started and stopped for the research since it was stated at the top of page 3 that individuals were studied between 2016 and 2018.  Also I do not understand why the age of the individual had to be reported using a scale.  Several of the individuals must have been the same age so reporting the actual age should not have decreased anonymity to a significant degree (especially if you did not use the actual birthdate).  It is also shown that several variables were changed to dichotomous variables but what these represent (which categories) was not given.  When you dichotomize in this fashion it helps makes the analyses more workable, however you lose information about the different gradients for the variables.  This should be noted.  For example, it is very different to say you smoke from time to time but not daily to more than 10 cigarettes daily.  If all the variables related to use were put in the “yes” category, information is certainly lost.  This also holds true for the injury severity questions.  Saying an injury is absent or non-absent oversimplifies the findings.  There is a great difference in injuries where the dancer may lose less than three days versus greater than seven days.

For the Results section, Figures 1A and 1B are not labeled as such on page 5.

In the Discussion Section, much information is provided about the role of cigarette smoking and alcohol consumption and level of professional experience, however, as stated earlier, more information should be provided about the possibility of other variables.  While the physiology surrounding the role of substances was interesting, I am not sure it needs to be provided in such depth at the exclusion of other discussion points.

On page 9, line 262, what do you mean by negative association?  This could imply an inverse association-do you mean that the greater the smoking the more pain and tendinopathy?  That would be positive and not negative-I am not clear here.

In the discussion section, the role of conducting face to face interviews, focus groups, or other qualitative methods should be emphasized more.  See my information said previously about socio-environmental factors that may be playing a very important role.

For more minor comments, the references do not appear to comply with the journal requirements-especially how to title the journals.  There were some grammatical errors in the manuscript.  For example, on page 8, lines 227-228, Advanced Training should be capitalized. On page 10, line 340, impact should be plural-impacts.  Finally on page 11, line 373, the word “on” should be changed to “of.”

Author Response

Dear Sir/Madam

Thank you for recognizing the potential of our work. Also, we are grateful for your elaborated and constructive critics and suggestions. We have tried to follow your comments and amended the manuscript accordingly. All changes are indicated specifically following each of your comments (please see RESPONSES)

Staying at your disposal!

Authors

REVIEWER 1

The manuscript, “Injury Occurrence and return to dance in professional ballet:  gender specific prospective analysis of specific correlates” adds to the injury prevention literature in the area of dance and especially ballet.  My major and minor comments are below.

My first comment has to do with the title of the manuscript.  While comparisons were made between males and females throughout the manuscript, I do not think it needs to be in the title as through most of the manuscript dancers as a whole were considered.

RESPONSE: The title is amended as suggested. It now reads: Injury occurrence and return to dance in professional ballet: prospective analysis of specific correlates

The Introduction provided an interesting overview of dance injuries and the lack of information available- especially in ballet.   While it is understandable why the variables such as substance use and smoking were reviewed, there was no information provided about what other variables might be considered in this research-including environmental variables that may play a role.  These variables might include socio-environmental stressors (need for constant achievement and excellence), lack of outside support, competition among dancers, etc.  These types of variables might best be analyzed through more qualitative techniques to help better explain why ballet dancers feel the need to use substances and smoke and perform other health-comprising behaviors.  While it is realized that not all variables could possibly be assessed, statements regarding other factors that may play a role should be provided.

RESPONSE: Thank you for your suggestion. Indeed, the identification of qualitative factors potentially related to injury occurrence is important topic in studying the ballet dance. Also, we were aware of the lack of our study with regard to this issue, and it was noted in the original version of the manuscript. In this version of the manuscript, the problem of qualitative study design is emphasized in the Conclusion section. Text reads: “The variables included in such qualitative investigations should not include only those observed herein. In brief, according to our results it is clear that due to need for constant achievement and excellence different socio-environmental stressors must be observed. Further, factors such as lack of outside support, variables explaining level of competition among dancers, etc. must be observed as well. Authors of the study believe that conducting studies which will allow identification of those factors (by face to face interviews, focus groups) will allow identification of different factors related not only to injury-, but also overall health-status and well-being of ballet dancers.” (please see last paragraph of the Conclusion).

For the Materials and Methods section, it was not clear when the year started and stopped for the research since it was stated at the top of page 3 that individuals were studied between 2016 and 2018.  Also I do not understand why the age of the individual had to be reported using a scale.  Several of the individuals must have been the same age so reporting the actual age should not have decreased anonymity to a significant degree (especially if you did not use the actual birthdate).  It is also shown that several variables were changed to dichotomous variables but what these represent (which categories) was not given.  When you dichotomize in this fashion it helps makes the analyses more workable, however you lose information about the different gradients for the variables.  This should be noted.  For example, it is very different to say you smoke from time to time but not daily to more than 10 cigarettes daily.  If all the variables related to use were put in the “yes” category, information is certainly lost.  This also holds true for the injury severity questions.  Saying an injury is absent or non-absent oversimplifies the findings. There is a great difference in injuries where the dancer may lose less than three days versus greater than seven days.

RESPONSE: Thank you for such detailed overview and explanation. There are responses to your suggestions.

·         With regard to “start of the year” by all means it should be noted. It is corrected (year always started with late August).

·         With regard to “exact age”… Indeed, it is hard to identify someone by exact age (because almost certainly several participants will have the same age of birth), but when combined with other variables (i.e. gender, performance level, troop) it becomes more possible. Therefore, and knowing the theoretical age-span in studied participants we decided to use categorization as in some previous studies.

·         With regard to “categorization” … Originally, we were aware of the possible “oversimplification” of the problems if variables will be categorized (as we did herein), but studies done so far regularly used such categorizations (even more “robust ones”), and therefore we followed previously established methodological approaches in studying the problem. By all means we will pay attention on these issues in our future studies, and try to avoid unnecessary dichotomisations whenever possible.

·         For a moment, the previous considerations are included in the Limitations sections, while some responses are incorporated into the Methods section. However, we must mention that Methods are systematically rewritten according to 2nd reviewer’s suggestions. Thank you!

For the Results section, Figures 1A and 1B are not labeled as such on page 5.

RESPONSE: The Figures are not included in this version of the manuscript as suggested by 2nd Reviewer.

In the Discussion Section, much information is provided about the role of cigarette smoking and alcohol consumption and level of professional experience, however, as stated earlier, more information should be provided about the possibility of other variables.  While the physiology surrounding the role of substances was interesting, I am not sure it needs to be provided in such depth at the exclusion of other discussion points.

RESPONSE: We agreed with your opinion. While we tried to explain all necessary details it turned out to be too detailed. Therefore, in this version of the manuscript the Discussion is significantly shortened (from 2500 to 1500 words) and extensive discussion on physiological background is avoided. Thank you

On page 9, line 262, what do you mean by negative association?  This could imply an inverse association-do you mean that the greater the smoking the more pain and tendinopathy?  That would be positive and not negative-I am not clear here.

RESPONSE: Thank you for your suggestion. It is amended now. Text reads: “Experimental studies confirmed correlation between smoking and shoulder pain and tendinopathy (the greater the smoking the more pain and tendinopathy) as well as increased risk of biceps tendon rupture”

In the discussion section, the role of conducting face to face interviews, focus groups, or other qualitative methods should be emphasized more.  See my information said previously about socio-environmental factors that may be playing a very important role.

RESPONSE: Thank you for this suggestion. While the discussion is focused solely on findings of our investigation, while we did not observe any qualitative variables, this issue is specifically overviewed in the Conclusion section. Text reads: “The variables included in such qualitative investigations should not include only those observed herein. In brief, according to our results it is clear that due to need for constant achievement and excellence different socio-environmental stressors must be observed. Further, factors such as lack of outside support, variables explaining level of competition among dancers, etc. must be observed as well. Authors of the study believe that conducting studies which will allow identification of those factors (by face to face interviews, focus groups) will allow identification of different factors related not only to injury-, but also overall health-status and well-being of ballet dancers.” (please see last paragraph of the Conclusion)

For more minor comments, the references do not appear to comply with the journal requirements-especially how to title the journals.  There were some grammatical errors in the manuscript.  For example, on page 8, lines 227-228, Advanced Training should be capitalized. On page 10, line 340, impact should be plural-impacts.  Finally on page 11, line 373, the word “on” should be changed to “of.”

RESPONSE: Thank you. All specified mistakes are corrected, and revised text is additionally checked for accuracy.

Staying at your disposal for any further changes and improvements

Authors

Reviewer 2 Report

Thank you for the opportunity to review the manuscript titled “Injury occurrence and return to dance in professional ballet: gender-specific prospective analysis of specific correlates”. This is relatively small but well-conceived prospective cohort study examining incidence, severity and risk factors of injury among professional ballet dancers. The manuscript is generally well-written, albeit unnecessarily verbose and cumbersome in some instances. I have a few comments and questions about the analysis, and I offer some suggestions to improve the reporting and readability of the manuscript.

ABSTRACT

Page 1, line 17: Remove hyphen in ‘injury-occurrence’.

Page 1, lines 23-24: Remove ‘observed by the Alcohol Use-Disorders-Identification-Test’.

INTRODUCTION

Page 1, line 34: Replace ‘i.e.’ with ‘e.g.’ as only a small selection of potential injuries are listed. Also, replace ‘sprain’ with ‘strain’.

Page 1, line 36: The hip is part of the lower extremity. Moreover, given that all major parts of the lower extremity are mentioned, there is really no need to list them. Simply say: ‘lower extremity and lower back’.

Page 1, lines 36-37: Add a summary of references to this statement. Also, remove the word ‘occurrence’, it is superfluous when taking about mechanism of injury.

Page 1, line 39: Remove the phrase ‘both male and female’, it is completely superfluous in this sentence.

Page 1, lines 42-43: Change to past perfect tense (i.e. ‘have reported’ and ‘have examined’). The phrase ‘prevalence of injury occurrence’ is potentially confusing, is it referring to prevalence or incidence of injury? Also, remove the phrase ‘occurrence as characteristic occupation risks’ as it serves no purpose in the sentence other than making it more difficult to understand. The juxtaposition of ‘dancers’ in the first part of the sentence with ‘professional ballet dancers’ in the second part is confusing. What ‘dancers’ does the first part of the sentence refer to? Non-ballet dancers? Non-professional ballet dancers? Please revise this sentence to make it clear.

Page 2, line 44: Were the ‘young dancers’ ballet dancers? Please clarify.

Page 2, line 46: Injury rate is a measure of risk; hence, there is no need to say ‘rates and risks’.

Page 2, lines 47-48: What is the difference between ‘hours of dance’ and ‘hours of exposure’? I would have thought that the exposure of interest here is (ballet) dance.

Page 2, line 55: From the subsequent discussion, it seems the interest is not only about injury prevalence, but injury more broadly (i.e. prevalence, incidence, severity, and risk factors). I suggest removing the word ‘prevalence’ here to avoid leading the reader to think the interest is confined to the prevalence of injury.

Page 2, line 76: Add the missing article ‘an’ before the word ‘important’.

Page 2, lines 81-83: These two sentences are somewhat unnecessary. Please consider replacing them by adding the something like following to the third sentence “Despite the significant injury problem in ballet, there is a lack of information on factors…” Furthermore, I think the implied but unstated element of the authors’ rationale is that the identification of risk factors will allow for developing potential intervention strategies to mitigate the injury problem. The authors may want to consider making this element more explicit in their rationale.

MATERIALS AND METHODS

Page 2, lines 92-93: So, 99 participants were included. But how many members were there in the National Ballet Theatres in Croatia and Slovenia during the study period? In other words, what is the size of the target population from which the sample of 99 is obtained? Also, were all members invited to participate? If not, then how many and what was the procedure for selecting who would be invited? On the basis of the size of the target population and/or number of invited participants, what was the response rate?

Page 3, line 111: In regard to sociodemographic factors, did the authors not record whether the participant belonged to the Croatian or Slovenian squad? I would imagine there could be multiple contextual factors that could be particular for each squad and potentially impact the injury risk.

Page 4, lines 144-154: The statistical analysis subsection needs further improvement. Firstly, there should be a mention of injury rates being calculated. Moreover, the injury rates should be calculated with 95%CI (for Poisson rates). Secondly, I think the authors should add a direct comparison of injury rates in males and females by calculating the rate ratio (with 95%CI). This would be the best method for determining if there is a significant difference in injury risk between males and females. I think this addition is particularly important given that the authors have decided to build separate regression models for each gender. Thirdly, I am curious about the authors’ rationale for collapsing the outcome measure to a binary variable (injured vs non-injured) in a logistic regression, when they could have retained more injury information by using Poisson regression. Lastly, irrespective of the type of regression used, the authors should describe the regression model building procedure (e.g. any backward or forward elimination) and comment on model diagnostics (e.g. issues with multicollinearity).

RESULTS

Page 4, lines 156-159: Replace ‘resulting in average of 1.95 injuries per dancer’ with ‘equating to an injury rate of 1.95 injuries per dancer per year’ (or alternatively, ‘per dancer-year’). Then, start a new sentence for the female vs male comparison. Add 95%CIs to the injury rates. Add the rate ratio with 95%CI to determine is the difference is significant.

Page 5, Figure 1: I am not convinced this figure adds any value and I think it should be removed. In particular, the proportions in panel 1a are already reported in the manuscript text and the graphical representation is unnecessary and unhelpful.

Page 5, lines 176-177: This sentence is yet another example of unnecessarily convoluted language. Please simplify to something like “Logistic regression did not reveal significant correlations between predictors and injury occurrence in males.”

Page 5, line 182: Replace the period after the OR with a comma.

Page 6, Table 1: Replace ‘F’ with the more conventional ‘n’ to represent frequencies. Also, consider adding a total row to help the reader see if there were any missing data.

Page 6, Figure 2: Again, this figure is unnecessary. (It also lacks an y-axis.) The relevant proportions are adequately reported in the manuscript text, although there appears to be some inconsistencies. In regard to the proportion of dances without absence, the text refers to 31% for males and 43% for females, whereas the figure suggests 31.7% (i.e. 32%) for males and 42.4% (i.e. 42%), respectively.

Page 6, Table 2: I would suggest adding to the table title a mention that these are the results from the logistic regression and the number of participants included in the regression (to help the reader see if there were any missing data).

Page 7, Table 3: As above, I would suggest adding to the table title a mention that these are the results from the logistic regression and the number of participants included in the regression (to help the reader see if there were any missing data).

DISCUSSION AND CONCLUSION

Although the discussion and conclusion sections are generally reasonable and balanced, their length (>2500 words) is very excessive. I recommend that the authors significantly reduce the length of these sections. By making the discussion and conclusion more succinct, the important findings will regain its emphasis and the readability of the manuscript will be much improved.

REFERENCES

For a relatively small and typical prospective cohort study, 55 references seems unnecessarily excessive. Reducing the length of the discussion should also help to reduce the number references.

Author Response

Dear Sir/Madam

First of all, we must say that this is probably one of the best and most constructive reviews we ever had. Thank you! We have tried to follow all your comments and suggestions and acted accordingly. Please see in the following text how we dealt with your comments.

Staying at your disposal for any further changes and improvements.

Authors

REVIEWER 2

Thank you for the opportunity to review the manuscript titled “Injury occurrence and return to dance in professional ballet: gender-specific prospective analysis of specific correlates”. This is relatively small but well-conceived prospective cohort study examining incidence, severity and risk factors of injury among professional ballet dancers. The manuscript is generally well-written, albeit unnecessarily verbose and cumbersome in some instances. I have a few comments and questions about the analysis, and I offer some suggestions to improve the reporting and readability of the manuscript.

RESPONSE: As we said, we are grateful for such detailed and elaborated comments and suggestions. We have tried to follow it specifically.

ABSTRACT

Page 1, line 17: Remove hyphen in ‘injury-occurrence’.

RESPONSE: Amended accordingly.

Page 1, lines 23-24: Remove ‘observed by the Alcohol Use-Disorders-Identification-Test’.

RESPONSE: Amended accordingly                   

INTRODUCTION

Page 1, line 34: Replace ‘i.e.’ with ‘e.g.’ as only a small selection of potential injuries are listed. Also, replace ‘sprain’ with ‘strain’.

RESPONSE: Amended accordingly

Page 1, line 36: The hip is part of the lower extremity. Moreover, given that all major parts of the lower extremity are mentioned, there is really no need to list them. Simply say: ‘lower extremity and lower back’.

RESPONSE: Amended accordingly.

Page 1, lines 36-37: Add a summary of references to this statement. Also, remove the word ‘occurrence’, it is superfluous when taking about mechanism of injury.

RESPONSE: Amended accordingly.

Page 1, line 39: Remove the phrase ‘both male and female’, it is completely superfluous in this sentence.

RESPONSE: Amended accordingly.

Page 1, lines 42-43: Change to past perfect tense (i.e. ‘have reported’ and ‘have examined’). The phrase ‘prevalence of injury occurrence’ is potentially confusing, is it referring to prevalence or incidence of injury? Also, remove the phrase ‘occurrence as characteristic occupation risks’ as it serves no purpose in the sentence other than making it more difficult to understand. The juxtaposition of ‘dancers’ in the first part of the sentence with ‘professional ballet dancers’ in the second part is confusing. What ‘dancers’ does the first part of the sentence refer to? Non-ballet dancers? Non-professional ballet dancers? Please revise this sentence to make it clear.

RESPONSE: Thank you for such detailed explanation. Text is amended and now reads: Previous studies have reported the injury occurrence and some factors associated with injury occurrence in ballet, but only few examined professional dancers” (please see beginning of the 2nd paragraph of the Introduction)

Page 2, line 44: Were the ‘young dancers’ ballet dancers? Please clarify.

RESPONSE: Indeed, this study observed young dancers from UK Centres for advanced training in dance, including ballet dance. It is now specified in the text.

Page 2, line 46: Injury rate is a measure of risk; hence, there is no need to say ‘rates and risks’.

RESPONSE: Amended accordingly, thank you.

Page 2, lines 47-48: What is the difference between ‘hours of dance’ and ‘hours of exposure’? I would have thought that the exposure of interest here is (ballet) dance.

RESPONSE: Indeed, it is a little bit confusing, but “dance exposures” is related to “single dance training no matter how long”. However, for better understanding the sentence now includes only “dance hours”. It reads now: “Australian authors prospectively examined injuries in pre-professional ballet dancers (15-19 years of age) and showed an injury rate of 1.38/1000 hours of dance”.

Page 2, line 55: From the subsequent discussion, it seems the interest is not only about injury prevalence, but injury more broadly (i.e. prevalence, incidence, severity, and risk factors). I suggest removing the word ‘prevalence’ here to avoid leading the reader to think the interest is confined to the prevalence of injury.

RESPONSE: Thank you for such elaborated comment. The word “prevalence” is removed.

Page 2, line 76: Add the missing article ‘an’ before the word ‘important’.

RESPONSE: Added, thank you!

Page 2, lines 81-83: These two sentences are somewhat unnecessary. Please consider replacing them by adding the something like following to the third sentence “Despite the significant injury problem in ballet, there is a lack of information on factors…” Furthermore, I think the implied but unstated element of the authors’ rationale is that the identification of risk factors will allow for developing potential intervention strategies to mitigate the injury problem. The authors may want to consider making this element more explicit in their rationale.

RESPONSE: By all means the development of the preventive strategy was the basic rationale of the study. The statement on it is included in the revised version of the manuscript (please see the last sentence of the Introduction)

MATERIALS AND METHODS

Page 2, lines 92-93: So, 99 participants were included. But how many members were there in the National Ballet Theatres in Croatia and Slovenia during the study period? In other words, what is the size of the target population from which the sample of 99 is obtained? Also, were all members invited to participate? If not, then how many and what was the procedure for selecting who would be invited? On the basis of the size of the target population and/or number of invited participants, what was the response rate?

RESPONSE More details on targeted population and response rates are included in the study now. Text now reads: “The participants in this one-year prospective study were 99 professional ballet dancers (41 males and 58 females). All participants were members of National Ballet Theaters from Croatia (n = 67; 39 females) and Slovenia (n = 32; 19 females) and were observed during the one-year period (starting from early September). The invitation for a participation was individually sent to all professionals of four ballet theatres (two of two, and two of four ballet theatres in Slovenia and Croatia, respectively). The target population was 175 dancers (70 and 85 in Slovenia and Croatia, respectively). Therefore, the response rate was 61% (43% and 79% for Slovenia and Croatia, respectively). Different ballet groups were observed during different years but all were tested between 2016 and 2018. During the course of the study, most of them participated in 25-30 dance hours per week in addition to performance…”

Page 3, line 111: In regard to sociodemographic factors, did the authors not record whether the participant belonged to the Croatian or Slovenian squad? I would imagine there could be multiple contextual factors that could be particular for each squad and potentially impact the injury risk.

RESPONSE: Actually, we were interested in these results also, but no significant differences between Slo and Cro squads were found, and therefore not reported.

Page 4, lines 144-154: The statistical analysis subsection needs further improvement. Firstly, there should be a mention of injury rates being calculated. Moreover, the injury rates should be calculated with 95%CI (for Poisson rates). Secondly, I think the authors should add a direct comparison of injury rates in males and females by calculating the rate ratio (with 95%CI). This would be the best method for determining if there is a significant difference in injury risk between males and females. I think this addition is particularly important given that the authors have decided to build separate regression models for each gender. Thirdly, I am curious about the authors’ rationale for collapsing the outcome measure to a binary variable (injured vs non-injured) in a logistic regression, when they could have retained more injury information by using Poisson regression. Lastly, irrespective of the type of regression used, the authors should describe the regression model building procedure (e.g. any backward or forward elimination) and comment on model diagnostics (e.g. issues with multicollinearity).

RESPONSE: Thank you for your suggestions. Please see responses bellow:

·         As you suggested, the calculation of injury rates (in injuries per 1000 hours of dance) is indicated in the Statistics section. Also, the injury rates were calculated with 95%CI, and differences between rates were calculated.

·         With regard to calculation of regressions… Indeed, the calculation of regression(s) with “non-dichotomized” outcomes, would be interested (and probably advantageous in some circumstances) but previous studies within the field regularly used such approach so we followed it. Also, there is one specific problem with regression-calculation using the “non-dichotomized” outcomes. Specifically, injury occurrence (outcome) is known to be strongly correlated to “previous injury”, and those dancers who suffered previous injury will have higher risk for “injury occurrence” (“outcome”). Therefore, if we will use “number of injuries” as an outcome for each participant, there is a certain possibility that “previous injury” will strongly influence “injury occurrence”. Consequently, “previous injury” will be a strong suppressor in regression calculation. This will probably result in lower (if any) influence of other predictors to outcome. The same “logic” is applicable for “injury severity” where previous injury is almost certainly strong predictor of “number of days-off” with the longer time-off for recidivists. Thank you!

·         With regard to regression model building… Actually, we calculated simple (i.e. univariate) logistic regression, and therefore collinearity between predictor variables was not the problem, and we didn’t check it prior to calculation. However, this is now clearly stated in the Statistics section. Thank you!

RESULTS

Page 4, lines 156-159: Replace ‘resulting in average of 1.95 injuries per dancer’ with ‘equating to an injury rate of 1.95 injuries per dancer per year’ (or alternatively, ‘per dancer-year’). Then, start a new sentence for the female vs male comparison. Add 95%CIs to the injury rates. Add the rate ratio with 95%CI to determine is the difference is significant.

RESPONSE: Thank you, it is amended as you specified. Text reads “During the one-year period, a total of 196 injuries occurred, equating to an injury rate of 1.9 injuries per dancer per year (95%CI: 1.6-2.3) . The higher frequency is founded in males than in females (2.4 [95%CI: 1.7-3.1] and 1.59 [95%CI: 1.1-2.1] injuries per dancer per year in males and females, respectively). It altogether corresponds with 1.4 (95%CI: 1.1-1.7) injuries per 1000 dance hours (1.7 [95%CI: 1.2-2.2] and 1.1 [95%CI: 0.7-1.5] injuries per 1000 HD in males and females, respectively). The percentage of dancers who reported no injury was similar across genders (23.7% and 25.42% for males and females, respectively), with no significant difference between genders (Chi square: 0.20, p = 0.66; OR: 1.55 [95%CI: 0.81-2.94]). Also, there was no significant difference between genders when number of injuries was observed on an ordinal scale (Mann-Whitney Z value: 1.11, p = 0.26).”

Page 5, Figure 1: I am not convinced this figure adds any value and I think it should be removed. In particular, the proportions in panel 1a are already reported in the manuscript text and the graphical representation is unnecessary and unhelpful.

RESPONSE: The Figure 1 is not included in the revised version of the manuscript. Thank you!

Page 5, lines 176-177: This sentence is yet another example of unnecessarily convoluted language. Please simplify to something like “Logistic regression did not reveal significant correlations between predictors and injury occurrence in males.”

RESPONSE: Corrected, thank you.

Page 5, line 182: Replace the period after the OR with a comma.

RESPONSE: Amended accordingly

Page 6, Table 1: Replace ‘F’ with the more conventional ‘n’ to represent frequencies. Also, consider adding a total row to help the reader see if there were any missing data.

RESPONSE: Corrected, thank you.

Page 6, Figure 2: Again, this figure is unnecessary. (It also lacks an y-axis.) The relevant proportions are adequately reported in the manuscript text, although there appears to be some inconsistencies. In regard to the proportion of dances without absence, the text refers to 31% for males and 43% for females, whereas the figure suggests 31.7% (i.e. 32%) for males and 42.4% (i.e. 42%), respectively.

RESPONSE: Amended accordingly. Specifically, the Figure 2 is not included in the revised version of the manuscript, while results are corrected. Thank you.

Page 6, Table 2: I would suggest adding to the table title a mention that these are the results from the logistic regression and the number of participants included in the regression (to help the reader see if there were any missing data).

RESPONSE: The title is changed accordingly, and now reads Results of the logistic regression… Also, number of participants is included in the table (please see first line in the table)

Page 7, Table 3: As above, I would suggest adding to the table title a mention that these are the results from the logistic regression and the number of participants included in the regression (to help the reader see if there were any missing data).

RESPONSE: Please see previous response.

DISCUSSION AND CONCLUSION

Although the discussion and conclusion sections are generally reasonable and balanced, their length (>2500 words) is very excessive. I recommend that the authors significantly reduce the length of these sections. By making the discussion and conclusion more succinct, the important findings will regain its emphasis and the readability of the manuscript will be much improved.

RESPONSE: Indeed, the discussion was long and therefore almost certainly difficult to read. In this version it is significantly shortened (from 2500 to about 1500 words). We believe that it improved the readability of the manuscript. Thank you.

REFERENCES

For a relatively small and typical prospective cohort study, 55 references seems unnecessarily excessive. Reducing the length of the discussion should also help to reduce the number references.

RESPONSE: Yes, reduction of the discussion reduced the number of references (from 55 to 44). Thank you.

Staying at your disposal!

Authors